# Genetic Variation of Physicochemical Properties and Digestibility of Foxtail Millet (*Setaria italica*) Landraces of Taiwan

**DOI:** 10.3390/molecules24234323

**Published:** 2019-11-26

**Authors:** Song-Yu Yin, Shu-Meng Kuo, Yu-Ru Chen, Yuan-Ching Tsai, Yong-Pei Wu, Yann-Rong Lin

**Affiliations:** 1Department of Agronomy, National Taiwan University, Taipei 10617, Taiwan; walter@almatec.com.tw (S.-Y.Y.); kuoshumeng@gmail.com (S.-M.K.); 2Crop Science Division, Taiwan Agricultural Research Institute, Taichung 41362, Taiwan; yuru0503@tari.gov.tw; 3Department of Agronomy, National Chiayi University, Chiayi 60004, Taiwan; yctsai@mail.ncyu.edu.tw; 4Department of Agronomy, Chiayi Agricultural Experiment Station, Taiwan Agricultural Research Institute, Chiayi 60044, Taiwan; wuypei@dns.caes.gov.tw

**Keywords:** foxtail millet, physicochemical properties, in vitro starch digestibility

## Abstract

Foxtail millet is considered a ‘smart food’ because of nutrient richness and resilience to environments. A diversity panel of 92 foxtail millet landraces preserved by Taiwan indigenous peoples containing amylose content (AC) in the range of 0.7% to 16.9% exhibited diverse physiochemical properties revealed by a rapid viscosity analyzer (RVA). AC was significantly correlated with 5 RVA parameters, and some RVA parameters were also highly correlated with one another. In comparison to rice, foxtail millet contained less starch (65.9–73.1%) and no significant difference in totals of resistant starch (RS), slowly digestible starch (SDS), hydrolysis index (HI), and expected glycemic index (eGI) according to in vitro digestibility assays of raw flour with similar AC. RS was significantly positively correlated with AC and four RVA parameters, cold paste viscosity (CPV), setback viscosity (SBV), peak time (PeT), and pasting temperature (PaT), implying that suitable food processing to alter physicochemical properties of foxtail millet might mitigate hyperglycemia. This investigation of pasting properties and digestibility of diverse foxtail millet germplasm revealed much variation and showed potential for multi-dimensional utilizations in daily staple food and food industries.

## 1. Introduction

Foxtail millet (*Setaria italica*) is considered one of the crops of the future because of its relatively high nutrient contents and great adaptability to environments, which provide resilience to climate change. Ranking second in the world production among millets, foxtail millet was domesticated around 6,000 BC and mainly cultivated as a staple food in southern Europe and temperate, subtropical, and tropical Asia [1,2]. In Taiwan, foxtail millet is a symbolic crop of the indigenous peoples and its cultivation dates to ~5,000 years ago [3]. Foxtail millet is a gluten-free and nutritionally superior whole grain containing various vitamins, minerals, and high levels of protein in comparison with major cereals such as rice and wheat [4]. These valuable traits make foxtail millet, along with other millets, ‘smart foods’, which offer multiple contributions to human well-being including food, fodder, fiber, nutrition, low environmental impact, and potential livelihood at minimal cost [5].

Being the major component of cereal grains, starch contributes to functional properties and nutritional characteristics of processed food products, such as infant food, pancakes, bread, and wine [6]. Starch is stored in the form of granules, which are classified as simple, compound, or semi-compound depending on granule number initiated in each amyloplast [7]. Foxtail millet starch granules were ~7.6 μm in diameter and mainly polygonal with a few spherical shapes [8]. The great majority of starch granules consists of amylose and amylopectin [6]. In foxtail millet grains, starch makes up 60–70% of dry matter, and amylose content (AC) ranges from 0 to 31.9% [4,9]. Foxtail millet grain was classified into three types based on AC, i.e., non-waxy, low-AC, and waxy. The non-waxy type was distributed worldwide, while the waxy type was distributed only in East Asia and Southeast Asia, accounting for almost half of the foxtail millet collection in Taiwan [9,10]. Starch viscosity is one of the key sensory parameters for cereal food processing, and can be assessed by the rapid viscosity analyzer (RVA), a useful tool for determining physicochemical properties to evaluate eating and cooking quality of cereal grains [11,12]. Parameters of starch physicochemical properties, such as peak viscosity (PKV), cold paste viscosity (CPV), setback viscosity (SBV), and peak time (PeT) were highly correlated with AC, and other factors were also suggested to influence pasting property [13,14].

The global prevalence of diabetes has increased by a remarkably high 54% in the past three decades due to population aging, transition of socioeconomic status, and changes of dietary habits [15]. Decreased risk of developing type 2 diabetes was associated with slowly digested and absorbed carbohydrates [16,17]. According to the time of digestion by in vitro enzymatic hydrolysis, starch can be fractionated into rapidly digestible starch (RDS), slowly digestible starch (SDS), and resistant starch (RS) [18]. SDS offered the advantage of a slow increase of postprandial blood glucose levels and sustained blood glucose levels over time compared to RDS [17]. High-AC rice varieties contain relatively high RS and low eGI, indicating the ratio of amylose and amylopectin was one of the main factors controlling digestibility [19].

Understanding starch properties would greatly support the improvement of sensory and textural properties of foxtail millet for uses in daily staple diet items and food industries. In Taiwan, 50.1% of landraces belong to the waxy type, and the investigation of traditional uses of foxtail millet from five ethnic groups revealed that waxy foxtail millets were for brewing and making dumplings, whereas non-waxy or low-AC grains were for porridge [10].

In this study, a diversity panel of foxtail millet landraces originating from Taiwan was evaluated to reveal starch granule structures and grain quality as reflected by aspects of AC and physicochemical properties by RVA, and in vitro digestibility along with the eGI value of grains. The results provide comprehensive and fundamental information about foxtail millet for promoting various applications in daily diets and commercial use in the food industry.

## 2. Results and Discussion

### 2.1. The Structure and Arrangements of Starch Granules in Foxtail Millet Grains

One accession from each of 3 different AC levels was selected for observing structure and arrangements of starch granules under SEM (Figure 1). The AC values of accession #387, #445 and SMCS-2 were 14.35%, 7.78% and 1.78%, respectively. The starch granules (S) of foxtail millet, which could be compound granules according to their polygonal shape and size, were tightly compacted in the endosperm (E) with few protein bodies (B) under 1500× magnification (Figure 1A,D,G). Starch granules ranging from 3 to 7 μm showed no obvious difference among AC levels (Figure 1C,F,I). Protein bodies, dimples resulting from protein bodies having fallen off during dissection, were commonly associated with starch granules in all three accessions. The cavity was found in waxy accessions but not in non-waxy and low AC accessions (Figure 1I,H).

Starch granules isolated from different botanical sources occur in various shapes, including spherical, polygonal, or oval [20]. Polygonal shape is commonly observed in cereal crops, such as sorghum, maize, oat and rice. The starch granule size of foxtail millets, proso millets, barnyard millets, kodo millets, and little millets were less than 10 μm in diameter with small spherical and small polygonal shapes [21]. Starch granules in foxtail millet were mainly polygonal with few spherical shapes, and the average sizes of starch granules of 53 Chinese foxtail millet varieties were in the range of 6.8 to 11.8 μm in diameter [22,23]. The sizes, structures, and arrangements of starch granules of foxtail millet estimated in this study were consistent with previous studies. Although the type of starch granules has not been verified, the starch granule of foxtail millet could be compound due to its polygonal shape and small size.

An internal cavity was evident in the starch granule of the waxy accession (Figure 1H,I). The cavity was identical to those in starch granules of sorghum, maize, and wild rice [24,25]. The surface pore and internal cavity in subfamily *Panicoideae*, including maize, sorghum, and millets, were a natural feature in starch granules [26]. The starch granule of rice with low AC showed more pores and cavities than the high AC counterparts. Starch granules in brown rice kernels with the same genetic background but different ACs all had cavities in their interiors [27]. These polygonal shape starch granules were intact in the nonchalky endosperm of kernels having high AC, which made the cavities hard to observe. However, cavities were clearly observed in the center of starch granules of low AC and waxy rice. The area of cavity was significantly negatively correlated with the AC. Starch granules having small cavities in high AC rice were difficult to break, while those with large cavities in low AC and waxy kernels were easily broken [27]. Besides, these pores and cavities may have osmotic effects during the starch gelatinization process because of facilitating water penetration and amylose leaching, directly leading to the swelling of starch granules [28]. Pores, channels, and cavities might be sites for enzyme attack that were capable of increasing the digestion rate and beneficial to fermentation [26,29]. These attributes of foxtail millet starch granules play an important role that should be taken into consideration in the modification of starch in the food industry; for example, the waxy type is suitable for brewing.

### 2.2. Physicochemical Properties of Foxtail Millet Grain

Starch from plants with different genetic backgrounds has diverse physicochemical properties that can be applied for different purposes, such as daily staple food, dessert, cracker and beverage production. AC has a strong relationship with pasting, thermal, and textural properties [14], thus is the main index by which to evaluate grain quality. The estimated AC of these 92 accessions of foxtail millet ranged from 0.7% to 16.9% (Table 1). According to a *Wx* genotyping assay, the AC of non-waxy genotypes ranged from 5.42% to 16.92%, which would be classified as low-AC types according to a previous study [9,10,30]. Consequently, in our study, the Q1 value of the AC of the non-waxy genotype was used to differentiate non-waxy and low-AC phenotypes. A total of 48, 21, and 23 accessions were classified as waxy, low-AC, and non-waxy types, respectively (Table 1). The AC of the waxy accessions ranged from 0.7% to 3.2% with a mean of 1.7%; the low-AC accessions ranged from 3.7% to 8.6% with a mean of 7.21%; the non-waxy accessions ranged from 8.7% to 16.9% with a mean of 12.07%. Non-waxy wild-type foxtail millets are distributed over Eurasia, East Asia, and South Asia; nevertheless, more than half of landraces collected from Taiwan are the waxy type because of indigenous peoples’ preferences. The low AC accessions of Nakayama’s collection could be found in Taiwan and Southeast Asia only; additionally, the non-waxy landraces from Taiwan and Southeast Asia showed slightly lower AC than those from South Asia and countries to the west [9,10]. The non-waxy type of foxtail millet is generally used as forage or for being directly cooked and consumed like rice, while the waxy type is often fermented for alcoholic beverages, or eaten in the form of sticky cake and foxtail millet dumplings [10]. Since grains with specific qualities are associated with human selection based on ethnological preferences, leading to various AC, efforts need to be intensified in understanding the pasting properties of foxtail millet starch to develop ideal varieties and products to meet multi-dimensional demands.

Pasting properties are important in the application of foxtail millet starch. The physicochemical property unveiled by RVA parameters exhibited high variation within this diversity panel of 92 accessions (Table 1, Figure 2). The frequency distributions of PKV, HPV, BDV, and PaT looked normal with few outliers. Nevertheless, the distributions of CPV and SBV were skewed to the right, and a bimodal distribution was observed in PeT (Figure 2).

AC was highly correlated with 5 parameters, with significant positive correlations with CPV (*R*^2^ = 0.81), SBV (*R*^2^ = 0.82), PeT (*R*^2^ = 0.59) and PaT (*R*^2^ = 0.57), but a significant negative correlation with HPV (*R*^2^ = −0.7) (Figure 2). The rice *Wx* locus, the major gene regulating AC, was responsible for HPV, BDV, CPV, and SBV in 3 segregating populations [13,31], and positively correlated with PKV, HPV, CPV, and SBV, but not correlated with BDV and P_time_ [14]. A significant positive correlation between AC and PaT was observed in foxtail millet herein (Figure 2) and in waxy and non-waxy rice because amylose interacts with amylopectin to form polymeric complexes in starch granules, resulting in hard and firm texture that requires more energy to dissolve [32] (Figure 2).

Correlation coefficients, 2-D scatter plot, and frequency distribution of RVA parameters and AC are presented on the upper, lower, and diagonal panel of matrix.

PKV had significant positive correlations with HPV (*R*^2^ = 0.32) and BDV (*R*^2^ = 0.94), but negative correlations with CPV (*R*^2^ = −0.28) and SBV (*R*^2^ = −0.34) (Figure 2). HPV showed significant negative correlations with CPV (*R*^2^ = −0.45), SBV (*R*^2^ = −0.59), PeT (*R*^2^ = −0.55), and PaT (*R*^2^ = −0.29). However, CPV showed significant positive correlations with SBV (*R*^2^ = 0.93), PeT (*R*^2^ = 0.43), and PaT (*R*^2^ = 0.56). SBV showed a significant positive correlation with PeT (*R*^2^ = 0.31) and PaT (*R*^2^ = 0.56), while PeT and PaT were also significantly positively correlated with each other (*R*^2^ = 0.46). The correlations among these parameters indicated that the pasting properties of foxtail millet starch provided reference information for food applications.

The distribution patterns of these RVA parameters were various among waxy, low-AC, and non-waxy types (Figure 2). Variations of RVA parameters in the same category were also observed -- apart from AC, the amylopectin structure, lipids, and residual proteins were also crucial factors affecting pasting property [33]. This phenomenon was found in 34 Chinese foxtail millet cultivars and breeding lines, whose AC narrowly ranged from 21.4% to 25.3%, but which exhibited highly diverse physicochemical properties [34].

Six RVA parameters showed a significant difference, except for PKV (Table 1; Figure 2). The non-waxy type had relatively low HPV and BDV, and high CPV, SBV, PeT, and PaT. In contrast, the waxy type had relatively high HPV and BDV, and low CPV, SBV, PeT, and PaT. The low-AC type showed intermediate CPV and SBV; however, its HPV and BDV were similar to the waxy type; and PeT and PaT were similar to the non-waxy type. In comparison with waxy accessions, low-AC accessions were expected to have lower PKV because they contained higher AC; however, the PKV of low-AC types was high in the present study. Low SBV is usually observed in waxy accessions, while high SBV usually displays a high retrogradation tendency. The retrogradation properties were determined largely by gelatinization temperature (GT) and the degree of crystallinity, which is affected by amylopectin chain-length distribution and AC [14]. Given that the structure of amylose is not as complicated as amylopectin, it is easier for amylose to form a gel network during retrogradation.

Specific applications of particular foxtail millet genotypes in diets and/or food processing might be suggested according to the evaluation of physicochemical properties, revealed by scatter plots of 92 accessions along 2 correlated RVA parameters (Figure 3). In general, waxy accessions had relatively low PaT (Figure 3A). High BDV in waxy accessions indicated that the starch is easy to be effectively broken down and modified by yeast, bacteria, or enzymes during fermentation, suggesting accession #480 would be a candidate for brewing. On the other hand, low BDV in non-waxy accessions indicates starch is stable and would not significantly influence other ingredients during food processing such as extrusion, suggesting accession #A280 would be a candidate for food processing. (Figure 3B). Low PaT and PeT are the two ideal characteristics for making millet porridge because of less energy consumption during cooking, suggesting that waxy accessions such as # SMCS-1 and #SMCS-2, have potential (Figure 3C). AC and CPV are two important factors that determine the eating and cooking quality [35]. Waxy accessions generally had relatively low AC and CPV; conversely, non-waxy accessions had relatively high AC and CPV (Figure 3D). Waxy grains are the preferred choice for the production of grain sorghum ethanol because the yield increases as the CPV decreases [36]. The high correlation between PKV and PaT along with the fine extraction after malting of barley suggested that low PaT and low AC are both suitable conditions for the malting process [37]. The preliminary investigation of physicochemical properties of this foxtail millet diversity panel paved the way for future breeding programs and application in daily staple food and food industries.

### 2.3. Starch Hydrolysis Analysis of Foxtail Millet

To mimic the starch digestibility in human digestive systems, the in vitro starch hydrolysis assay was performed on 9 foxtail millet accessions and 3 rice varieties with various AC. In rice, the AC of IR8, TN11, and Hung-No were 20.7%, 12.2%, and 1.4%, respectively. For foxtail millet, three non-waxy accessions with 14.4–16.9% AC, two low-AC accessions with 6.0–7.7% AC, and 4 waxy accessions with 1.3–1.9% AC were selected. The rice flour in this study was made of polished white rice, while the foxtail millet flour was made of dehusked but unpolished grains because polishing is not a required process for utilization of foxtail millet. The starch content of foxtail millet ranged from 65.9% to 73.1%, while those of three rice cultivars ranged from 88.9 to 91.1% (Table 2). Starch contents of rice varieties were significantly higher than foxtail millets regardless of AC levels, consistent with previous studies [4,38].

The starch digestibility of grain flours was assessed by in vitro enzymatic hydrolysis lasting 120 min (Appendix A). In general, the hydrolysis rates were negatively correlated with AC. The hydrolysis curves showed that *indica* rice with the highest AC, IR8, had the slowest hydrolysis rate, indicating the least digestible starch. The hydrolysis rate of glutinous rice, Hung-No, was higher than japonica rice, TN11. In foxtail millet, the hydrolysis rate from highest to lowest was waxy, low-AC, and non-waxy. Also, the hydrolysis rates of all examined foxtail millets were higher than the 3 rice varieties, indicating that foxtail millet starch was relatively easy to digest.

Starch was classified into RDS, SDS, and RS according to the time of digestion [18]. In the non-waxy foxtail millet accessions, the proportions of the 3 starch fractions were RDS (13.1–20.9%) < SDS (30.6–42.2%) < RS (41.7–51.2%) (Table 2). The low-AC foxtail millet showed a similar tendency: RDS (17.1–17.6%) < SDS (32.2–33.8%) < RS (48.6–50.7%). Nevertheless, the proportions in the waxy type was different from others: RDS (20.2–26.8%) < SDS (36.3–43.1%) ≈ RS (35.2–40.7%). The proportion of 3 starch fractions varied among different accessions, even with similar AC. In general, RS was positively correlated with AC while RDS showed the opposite relationship (Table 2). Waxy foxtail millet possessed higher RDS than the non-waxy and low-AC types, and this pattern was also found in rice -- IR8 with high AC had the lowest proportion of RDS (11.2%) and the highest proportion of RS (67.9%). Within a similar AC, foxtail millet showed higher SDS but lower RS than rice, while no significant difference in RDS was found in these two species. The digestibility of foxtail millet is often considered to be lower than rice due to its high fiber content and protein availability. However, the samples used in this experiment were raw materials whose starch granules had not gone through complete gelatinization; thus, some structure of starch granules remained intact. Besides, surface pores and internal cavities in foxtail millet starch granules allow enzymes to access the whole starch granule rapidly. These features have not been found in rice starch, which may result in a relatively low digestion rate [26,39] (Figure 1).

In foxtail millet, high eGI were observed in waxy accessions, while no significant difference was detected between non-waxy and low-AC accessions. In rice, the eGI of Hung-No was also significantly higher than TN11 followed by IR8, while the latter two varieties showed no significant difference. Raw foxtail millet starch possessed higher HI and eGI than rice in all AC levels, indicating that foxtail millet starch was easier to digest than rice though they contained less starch (Table 2). The characteristics of millet starch may not be the main factor affecting the hypoglycemic property -- lipids, proteins, and phenolic compounds might be contributing factors [40]. For instance, the content of RDS in raw materials was millet flour (37.7%) < defatted millet flour (40.1%) < deproteinated millet flour (50.9%) < millet starch (53.4%), whereas the content of SDS and RS showed converse patterns [41].

The starch property of foxtail millet-based food was significantly different under boiling, steaming, and extrusion preparations; for example, high eGI can be found in millet porridge, followed by millet pancake, and cooked millet [41]. The starch digestibility of millet flour was significantly lower than that of wheat flour both under raw and cooked conditions [41]. Even raw foxtail millet had higher GI than rice in this study (Table 2) — we noticed that millets and millet-based products had lower glycemic and insulinemic response compared to other cereals [42]. For example, biscuits made from foxtail millet flour had the lowest GI of 50.8 compared to 68 for biscuits from barnyard millet flour and refined wheat flour [40]. A significant reduction (*p* < 0.001) was observed in the postprandial glucose level of patients who consumed a millet-based dosa when compared to those who consumed a rice-based dosa [43]. Millet-based products might play a protective role in the management of hyperglycaemia [43]; moreover, millets without gluten make them suitable for people with celiac disease.

The digestibility of starch is diverse and influenced by particle size, composition, physicochemical properties, food processing conditions, and plant types [44]. The correlation between starch fractions and physicochemical properties of foxtail millets was analyzed (Table 3). AC was significantly positively correlated with RS (*R*^2^ = 0.59) but not significantly correlated with RDS and SDS. From the perspective of the physicochemical property of amylose, long-linear chains characteristic in starch granules interact with amylopectin and lipid, and hence amylose is resistant to enzyme digestion [45]. RS content showed a significant positive correlation between CPV, SBV, PeT, and PaT with correlation coefficients of 0.67, 0.64, 0.82 and 0.63, respectively. RDS showed a significant negative correlation between these four parameters with correlation coefficients of −0.56, −0.56, −0.78 and −0.64, respectively (Table 3). The relationship of AC, enzyme digestion, and pasting in raw and cooked rice both showed similar results as the foxtail millet starch in this study [46,47]. As a result, AC, CPV, SBV, PeT, and PaT could be used as indices corresponding to RS and RDS content.

Decreased risk of developing type 2 diabetes was associated with slowly digested and absorbed carbohydrates [16,17]. In our study, *indica* rice, IR8, exhibited the highest RS and lowest HI and eGI because of high AC (>20%) (Table 2), and many rice accessions possessing more than 20% AC showed low GI as well [48]. However, the AC of the collected foxtail millet landraces from Taiwan was relatively low, <16.9%, due to indigenous peoples’ preferences. No significant difference in the SDS, RS, HI, and eGI between *japonica* rice TN11 and non-waxy foxtail millets was observed (Table 2). Physiochemical properties which would be altered during food processing affect digestibility the starch gelatinization was a process turning SDS and RS into RDS; also, millet and millet-based products are known to have lower starch and protein digestibility rates [40]. We noticed that RS was significantly positively correlated with AC, and four RVA parameters (Table 3); thus, appropriately processed foxtail millet products with a low GI have the potential to mitigate hyperglycemia. Future studies on the current topic are therefore needed to investigate the digestibility of food products derived from the materials in this study.

## 3. Materials and Methods

### 3.1. Plant Materials and Experimental Design

A diversity panel of 92 foxtail millet accessions was evaluated [10]. (Appendix A). All accessions were cultivated in the net hut of Taiwan Agricultural Research Institute (TARI), Taichung, Taiwan, in a randomized complete block design (RCBD) with three replications in the fall of 2015. For the in vitro starch digestibility assay, three rice (*Oryza sativa*) varieties, IR8 (subspecies *indica*), TN11 (*japonica*), and Hung-no (*indica* glutinous cultivar), were included for comparison. IR8, developed by the International Rice Research Institute (IRRI), is a semi-dwarf and high-yielding rice variety widely considered to have played an important role in the rice Green Revolution; TN11 is a leading variety in Taiwan. Rice varieties were grown in TARI Chiayi branch in a RCBD with two replicates in 2015.

### 3.2. Scanning Electron Microscopy (SEM) of Starch Granules

Foxtail millet grain samples were dried at 37 °C for a week. Dehusked grains were transversely cut and fixed on the specimen holder by carbon conductive double-faced adhesive tape. The grain surfaces were gold coated in an ion sputter E-1010 (Hitachi, Japan) for 60 seconds at 2.4 kV, and images of grain surfaces were taken at 1500, 3000, and 6000× magnification by using FEI Quanta 200 (Hillsboro, OR, USA).

### 3.3. Amylose Content

AC was determined by the iodine-staining (I_2_-KI) assay with minor modification [49]. The amylose standard solution contained 40 mg of potato amylose (Sigma, USA), 50 μL of 95% ethanol and 450 μL of 1 M NaOH in 1.5 mL of total volume and was diluted to 0%, 10%, 20%, 30%, and 40% to set up a standard curve. Foxtail millet grains were dehusked, ground into fine flour, and sieved through a 0.43 mm mesh (40 Mesh). Each sample was prepared by following the same procedure as the standard solution. A total of 5 μL sample was then mixed with 295 μL staining solution prepared by mixing 5 mL of H_2_O, 0.5 mL of 1 M acetic acid and 0.5 mL of iodine solution (2 % KI and 0.2 % I_2_). The AC of each sample was calculated according to the standard curve based on the absorption at O.D. 620 nm (U-3010 UV-Visible Spectrometer, Hitachi, Japan).

### 3.4. Pasting Properties of Foxtail Millet Grain

The viscosity of cooked foxtail millet grain was analyzed using the Rapid Visco Analyzer (Model No. RVA-4; Newport Scientific, Australia) to obtain RVA profiles. Dehusked foxtail millet grains were ground into fine flour and sieved through a 0.43 mm mesh (40 Mesh). Approximately 3 g of foxtail millet flour was mixed with 25 mL of water in an RVA sample canister. The sequential temperature curve for 23 min was as follows: (1) holding at 50 °C for 10 sec at 960 rpm and then incubating at 50 °C for 1 min; (2) heating to 95 °C and holding for 3.7 min; (3) cooling to 50 °C and holding at 50 °C for 3 min (Appendix A). Seven RVA parameters, the peak viscosity (PKV), hot paste viscosity (HPV), cool paste viscosity (CPV), pasting temperature (PaT), peak time (PeT), and two derivative parameters, breakdown viscosity (BDV = PKV − HPV), and setback viscosity (SBV = CPV − PKV) were recorded.

### 3.5. Starch Content

The total starch content of foxtail millet was determined by using Megazyme starch assay kit (Megazyme International, Wicklow, Ireland) according to AOAC Method 996.11 (AACC Method 76-13.01). The total glucose content was determined by the d-Glucose Assay Kit (GOPOD Format, Megazyme International, Ireland) and then converted to starch content.

### 3.6. In Vitro Starch Digestibility

The flour of two to four foxtail millet accessions from each AC level was analyzed by using in vitro starch hydrolysis based on Englyst et al. (1999) with some modifications. In rice, IR8, TN11, and Hung-No, representing high AC, intermediate AC, and waxy type were included for comparison. Approximately 0.5 g flour was dispersed in 10.0 mL of freshly prepared pepsin solution (50 mg of pepsin and 50 of guar gum in 10 mL of 0.05 M HCl) and incubated at 37 °C for 30 min at 200 rpm. Subsequently, 10.0 mL of 0.1 M acetate buffer (pH 5.5, 37 °C) was added to form a buffer at pH 5.2. The enzyme mixture was prepared by dispersing 3.0 g of pancreatin in 20.0 mL of H_2_O in the tube, shaking for 10 min, and then centrifuging at 1500 g for 10 min; 15.0 mL of pancreatin supernatant was saved, subsequently adding 0.75 mL of amyloglucosidase (1200 U mL^−1^) and 1 mL of invertase (3000 U mL^−1^). Starch digestion was initiated by adding 5 mL of the enzyme mixture and then incubating at 37 °C for 3 h under horizontal shaking at 200 rpm. During the incubation, 0.5 mL of each sample was added into 10 mL of absolute ethanol and mixed well every 20 min. These digested samples were collected at 7 time points, specifically 20 (G**_20_**), 40, 60, 80, 100, 120 (G**_120_**), and 180 min (TG), and the supernatant was saved after centrifugation at 1500× *g* for 5 min. The glucose portion in the supernatant was measured according to the glucose oxidase-peroxidase method by using a GOD-POD diagnostic kit (Applygen Technologies, Beijing, China). The content of three starch fractions, RDS, SDS, and RS was calculated on a dry basis with the following equations and expressed as percentage of total starch [50]:(1)$$\mathrm{RDS}=G_{20}\times 0.9$$

(2)$$\mathrm{SDS}=\left(G_{120}-G_{20}\right)\times 0.9\;$$

(3)$$\mathrm{RS}=\left(TG-\;G_{120}\right)\times 0.9\;$$

The estimated glycemic index (eGI) was calculated based on glucose measurements at 7 time points mentioned above during starch hydrolysis. The kinetics of in vitro starch digestion was calculated according to the first order equation [51]:(4)$$C=C\infty \left(1-e^{-kt}\right)$$
where C and *t* correspond to the percentage of starch hydrolyzed at *t* time. C∞ is the equilibrium concentration and *k* is the kinetic constant in *t* time. Area under curve (AUC) was obtained each sample by using the equation:(5)AUC=C∞(tf−t0)−(C∞/k)[1−exp[−k(tf−t0)]]
where *tf* and *t0* are the final (180 min) and initiate (0 min) hydrolysis time, respectively.

The hydrolysis index (HI) was calculated by dividing the area under curve (AUC) value of each sample with the AUC of the reference food, *japonica* rice TN11. The expected (eGI) was converted by following equation:(6)$$\mathrm{eGI}=39.21+0.803\left(H_{100}\right)$$
which *H*_100_ is the glucose content measured at 100 min.

### 3.7. Statistical Analysis

AC and the starch hydrolysis parameters were estimated for 3 samples and expressed as mean ± SD. For AC, RVA, and starch hydrolysis parameters, one-way ANOVA was performed to investigate statistical significance of differences between parameters, followed by Fisher’s LSD. Pearson correlation coefficients between variables were calculated with a significance threshold of *p* < 0.05.

## Figures and Tables

**Figure 1 molecules-24-04323-f001:**
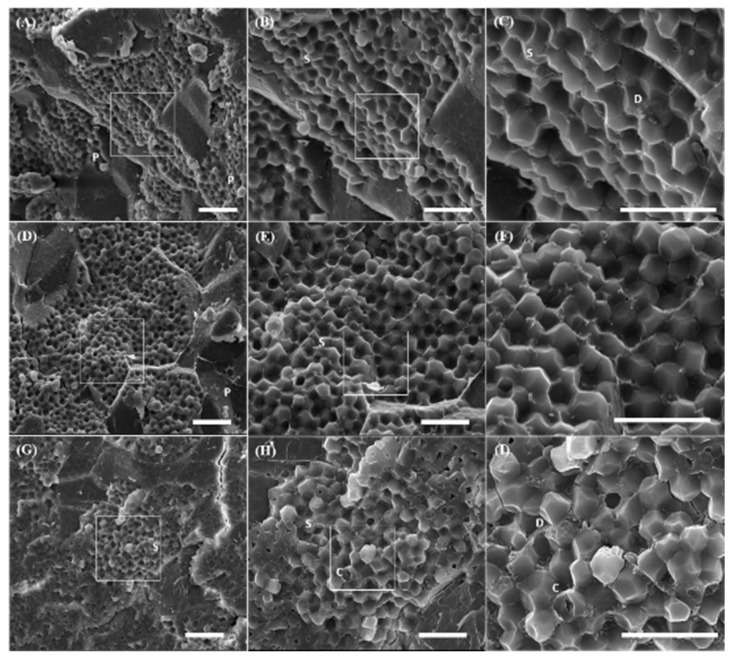
Starch granules in transverse sections of foxtail millet grains. The grains of three accessions with different AC levels, non-waxy (**A**–**C**), low-AC (**D**–**F**) and waxy (**G**–**I**), were observed under 1500×, 3000×, and 6000× magnification. Regions indicated by the squares were magnified in the figure on the right side. C, cavity; D, dimple; P, protein body; S, starch granule. Bars = 20 μm.

**Figure 2 molecules-24-04323-f002:**
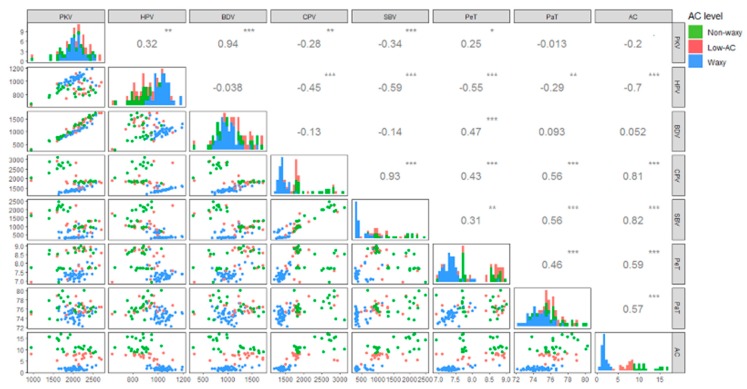
Correlation matrix of physicochemical property characters.

**Figure 3 molecules-24-04323-f003:**
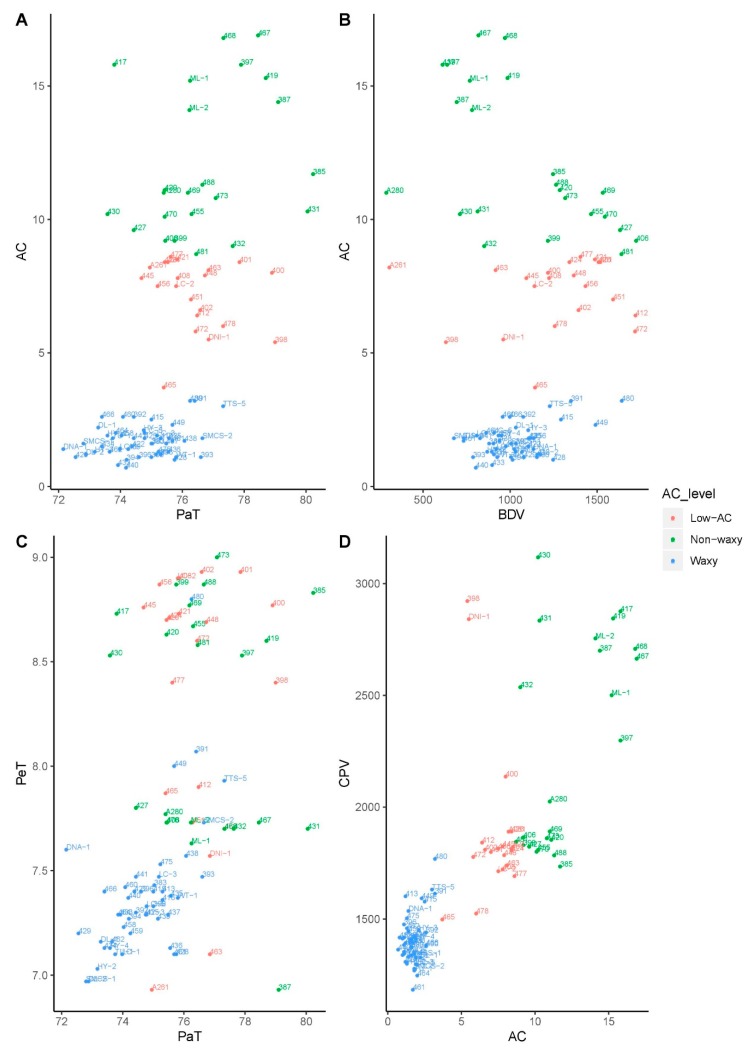
2D scatter plots of 92 foxtail millet accessions on two different physicochemical properties. The relationships between AC and PaT (**A**), AC and BDV (**B**), PeT and PaT (**C**), and CPV and AC (**D**) are represented by three AC levels.

**Table 1 molecules-24-04323-t001:** Physicochemical properties of foxtail millet grains.

		Endosperm phenotype		
Trait		Waxy	Low-AC	Non-waxy
No. of accessions		48	21	23
AC (%)	Range	0.7–3.2	3.7–8.6	8.7–16.9
	Mean	1.7 ^c^	7.21 ^b^	12.07 ^a^
PKV (cP)	Range	1641.5–2574.5	940–2706	942–2555
	Mean	2051.76 ^a^	2143.60 ^a^	1931.32 ^a^
HPV (cP)	Range	908-1190	634–1115	654–1023
	Mean	1033.17 ^a^	888.0 ^a^	853.25 ^b^
BDV (cP)	Range	677–1641.5	306–1720.5	288.5–1724
	Mean	1018.59 ^ab^	1255.6 ^a^	1078.07 ^b^
CPV (cP)	Range	1183.5–1768	1498–2922	1734–3117.5
	Mean	1403.75 ^c^	1888.35 ^b^	2258.76 ^a^
SBV (cP)	Range	291–1136.5	476–2243.5	682.5–2491
	Mean	427.2 ^c^	1108.8 ^b^	1555.99 ^a^
PaT(℃)	Range	72.15–77.32	74.68–79	73.58–80.23
	Mean	74.67 ^b^	76.36 ^a^	76.69 ^a^
PeT(min)	Range	6.93–9	6.93–8.93	6.97–8.8
	Mean	7.37 ^b^	8.34 ^a^	8.21 ^a^

Means with different letter superscripts are significantly different at *p* < 0.05 according to Fisher’s LSD test. AC, apparent amylose content; PKV, peak viscosity; HPV, hot past viscosity; BDV, breakdown viscosity; CPV, cool past viscosity; SBV, setback viscosity; PeT, peak time; PaT, pasting temperature. Centipoise (cP) is a physical unit defining the dynamic viscosity, which 1cP equals to 0.001 N·s·m^−2^.

**Table 2 molecules-24-04323-t002:** Summary of in vitro starch digestibility.

Accession^*^	AC (%)	Starch Content (%)	RDS (%)	SDS (%)	RS (%)	HI	eGI
387	14.4 ± 3.6 ^bc^	72.4 ± 1.3 ^b^	20.9 ± 1.8 ^bc^	37.4 ± 9.0 ^abc^	41.7 ± 7.3 ^bcd^	119.7 ± 15.4 ^cd^	104.9 ± 8.4 ^cd^
419	16.0 ± 1.1 ^b^	67.4 ± 1.7 ^b^	18.2 ± 2.5 ^bcd^	30.6 ± 3.3 ^bcd^	51.2 ± 2.1 ^d^	124.6 ± 3.1 ^cd^	107.6 ± 1.7 ^cd^
467	16.9 ± 0.2 ^ab^	73.1 ± 4.2 ^b^	13.1 ± 1.1 ^de^	42.2 ± 2.8 ^ab^	44.7 ± 1.8 ^bcd^	137.8 ± 5.5 ^bc^	114.9 ± 3.0 ^bc^
445	7.7 ± 2.0 ^d^	71.1 ± 2.2 ^b^	17.1 ± 1.0 ^bcd^	32.2 ± 3.4 ^abcd^	50.7 ± 2.6 ^b^	141.2 ± 17.3 ^abc^	116.7 ± 9.5 ^abc^
478	6.0 ± 0.7 ^d^	69.9 ± 4.4 ^b^	17.6 ± 1.1 ^bcd^	33.8 ± 0.1 ^abc^	48.6 ± 1.2 ^bc^	134.9 ± 5.6 ^bc^	113.3 ± 3.1 ^bc^
434	1.5 ± 0.4 ^e^	67.6 ± 4.7 ^b^	26.8 ± 1.9 ^a^	36.3 ± 4.3 ^abc^	36.8 ± 3.6 ^d^	165.9 ± 7.0 ^a^	130.3 ± 3.9 ^a^
436	1.3 ± 0.1 ^e^	70.6 ± 3.7 ^b^	20.8 ± 0.4 ^bc^	38.5 ± 0.5 ^abc^	40.7 ± 0.4 ^cd^	158.5 ± 6.9 ^ab^	126.2 ± 3.8 ^ab^
HY4	1.9 ± 0.8 ^e^	71.4 ± 2.5 ^b^	20.2 ± 1.4 ^bc^	42.7 ± 2.8 ^a^	37.1 ± 1.7 ^d^	154.9 ± 0.4 ^ab^	124.3 ± 0.2 ^ab^
LC3	1.9 ± 0.0 ^e^	65.9 ± 0.8 ^b^	21.7 ± 2.1 ^ab^	43.1 ± 3.9 ^a^	35.2 ± 4.9 ^d^	164.9 ± 5.8 ^a^	129.7 ± 3.2 ^a^
IR8	20.7 ± 0.7 ^a^	90.5 ± 2.6 ^a^	11.2 ± 0.6 ^e^	21.0 ± 2.6 ^d^	67.9 ± 2.5 ^a^	77.8 ± 6.2 ^e^	82.4 ± 3.4 ^e^
TN11	12.2 ± 0.9 ^c^	91.1 ± 3.0 ^a^	18.0 ± 1.3 ^cd^	24.2 ± 0.7 ^d^	57.8 ± 0.5 ^a^	100.0 ± 0.0 ^de^	94.6 ± 0.0 ^de^
Hung-No	1.4 ± 0.2 ^e^	88.9 ± 2.3 ^a^	20.4 ± 2.7 ^bc^	27.3 ± 3.6 ^cd^	52.3 ± 0.9 ^b^	112.6 ± 3.5 ^c^	101.5 ± 1.9 ^c^

^*^ Nine foxtail millet accessions chosen from three AC levels; IR8, TN11, and Hung-No are rice cultivars representing high AC, intermediate AC, and waxy type, respectively. Three replicates were executed for each accession for each assay. Means with different superscripts are significantly different at *p* < 5% according to Fisher’s LSD test. RDS, rapidly digestible starch; SDS, slowly digestible starch, RS, resistant starch; AC, amylose content; HI, hydrolysis index; eGI, expected glycemic index.

**Table 3 molecules-24-04323-t003:** Correlation between starch fractions and physicochemical properties of foxtail millets.

Parameter	RDS	SDS	RS
Starch content	−0.50	−0.15	0.38
AC	−0.55	−0.40	0.59 *
PKV	−0.13	−0.47	0.41
HPV	0.02	−0.23	0.16
BDV	−0.15	−0.47	0.42
CPV	−0.56 *	−0.50	0.67 *
SBV	−0.56 *	−0.46	0.64 *
PeT	−0.78 **	−0.54	0.82 ***
PaT	−0.64 *	−0.39	0.63 *

* *p* < 0.05, ** *p* < 0.01, *** *p* < 0.001, RDS, rapidly digestible starch; SDS, slowly digestible starch, RS, resistant starch; AC, apparent amylose content; PKV, peak viscosity; HPV, hot paste viscosity; BDV, breakdown viscosity; CPV, cool paste viscosity; SBV, setback viscosity; PeT, peak time; PaT, pasting temperature.

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
