# Peer review of "Genetic Variation of Physicochemical Properties and Digestibility of Foxtail Millet (Setaria italica) Landraces of Taiwan"

_molecules, 2019, doi:10.3390/molecules24234323_

Round 1
Reviewer 1 Report
The authors investigated the morphology, amylose content, pasting properties and digestibility of foxtail millet landraces of Taiwan. I have some comments as the below.
Title. The authors only investigated the amylose content and pasting properties, and it is improper to call them as physicochemical properties. The title needs to be changed. It might be more proper for “Morphology, amylose content, pasting properties and digestibility of foxtail millet landraces of Taiwan. It is proper to use the “pasting properties” instead of “physicochemical properties” for the results determined by RVA in the whole manuscript. L96-115. Recently, Zhang et al. (J. Cereal Sci. 2019, 90: 102854) report the relationships between starch cavity and transparency, amylose content and moisture of brown rice kernels. It is better to give more discussion according to the references. L107. It is better to use the scale bar instead of the magnification due to that the photographs were scaling when inserted in the text. The quality of Fig. 3 needs to be improved. L356. Please check the 40 mm mesh. Is it 40 μm?Author Response
Please see the attachment

Reviewer 2 Report
I reviewed an article entitled: “Genetic Variation of Physicochemical Properties and Digestibility of Foxtail Millet (Setaria italica) Landraces of Taiwan”. This work details about the starch characteristics of foxtail millet varieties in terms of amylose content, pasting properties, starch digestibility classification and estimated glycemic index. This work used many foxtail millet varieties which is appreciated. However, the data is of only descriptive in that it only tells about the starch properties of the flours which has little to No importance to predict the digestibility based starch properties and glycemic index of such starch after cooking. Cooking significantly changes the properties of starch in that only by looking at the starch properties of flour, it is not possible to give any prediction how such starches can behave after cooking processing. Therefore, to complete or give a full story, it would have been good if the authors have seen how the digestibility based starch classification and GI are affected by certain cooking and see if the starch properties of the starting material has an effect on the starch properties of cooked food products. The other concern I have is that I couldn’t see distinctive methods how the digestibility based starch classification and GI are determined.
Round 2
Reviewer 1 Report
The authors have revised their manuscript according to my comments. I recommend accepting it for publication.